

# Social Motor Priming: when offline interference facilitates motor execution

Sonia Betti[1], Eris Chinellato[2], Silvia Guerra[1], Umberto Castiello[1] and Luisa Sartori[1,3]

[1] Dipartimento di Psicologia Generale, Università degli Studi di Padova, Padova, Italy
[2] Department of Design Engineering and Mathematics, Middlesex University, London, United Kingdom
[3] Padova Neuroscience Center, Università degli Studi di Padova, Padova, Italy

## ABSTRACT

Many daily activities involve synchronizing with other people's actions. Previous literature has revealed that a slowdown of performance occurs whenever the action to be carried out is different to the one observed (i.e., visuomotor interference). However, action execution can be facilitated by observing a different action if it calls for an interactive gesture (i.e., social motor priming). The aim of this study is to investigate the costs and benefits of spontaneously processing a social response and then executing the same or a different action. Participants performed two different types of grips, which could be either congruent or not with the socially appropriate response and with the observed action. In particular, participants performed a precision grip (PG; thumb-index fingers opposition) or a whole-hand grasp (WHG; fingers-palm opposition) after observing videos showing an actor performing a PG and addressing them (interactive condition) or not (non-interactive condition). Crucially, in the interactive condition, the most appropriate response was a WHG, but in 50 percent of trials participants were asked to perform a PG. This procedure allowed us to measure both the facilitator effect of performing an action appropriate to the social context (WHG)—but different with respect to the observed one (PG)—and the cost of inhibiting it. These effects were measured by means of 3-D kinematical analysis of movement. Results show that, in terms of reaction time and movement time, the interactive request facilitated (i.e., speeded) the socially appropriate action (WHG), whereas interfered with (i.e., delayed) a different action (PG), although observed actions were always PGs. This interference also manifested with an increase of maximum grip aperture, which seemingly reflects the concurrent representation of the socially appropriate response. Overall, these findings extend previous research by revealing that physically incongruent action representations can be integrated into a single action plan even during an offline task and without any training.

## INTRODUCTION

A wealth of research has shown that motor response is facilitated (e.g., shorter reaction times and increased accuracy) if the action to be carried out has been recently observed. This phenomenon, termed visuomotor priming, has been defined as the facilitation to perform an action congruent with the observed one (*Heyes, 2011*). Interestingly, visuomotor

Corresponding author
Luisa Sartori, luisa.sartori@unipd.it

priming can be reversed when others' actions are instrumental for the fulfillment of a specific joint goal. If, for example, someone holding a mug by its handle—using a three-digit grasp—hands it to us, we automatically grasp the 'available' surface with a whole-hand grip, rather than imitating the observed grip. In this case, the two actions are physically incongruent, yet complementary. Recent findings have shown that the brain can easily resolve the conflict between the automatic tendency to imitate, and the need to perform context-related complementary actions (*Sacheli et al., 2015*; *Era et al., 2018a*; *Era et al., 2018b*; for a review see *Sartori & Betti, 2015*). Brief periods of sensorimotor experience, in which participants are trained to perform a different action from the one observed, can even abolish (*Cook et al., 2010*) or reverse (*Catmur et al., 2008*) visuomotor priming. It then appears that action observation does not automatically lead to imitation, but rather, depending on sensorimotor experience, observed actions could prime different responses.

Learnt social responses can modulate motor performance (*Liepelt, Prinz & Brass, 2010*; *Flach et al., 2010*), so that complementary response preparation can spontaneously overwhelm imitative responses (i.e., social motor priming; *Wang & Hamilton, 2013*; *Hamilton, 2013*). *Liepelt, Prinz & Brass (2010)* described the *reversed compatibility* effect when observing a human right hand extended for a handshake: participants responded faster with their own right hand, not with their mirror hand (*Liepelt, Prinz & Brass, 2010*). This effect is driven by the strongly life-long learnt social response of responding with the non-mirror hand to handshaking. Continuing on this analysis, also prosocial words (e.g., "join") shown prior to action observation are able to shape perception and behavior (*Leighton et al., 2010*). Of relevance for the present study, verbal social primes can increase interference when participants execute an action different from the observed one (*Sparks, Douglas & Kritikos, 2016*). Along with visuomotor effects, visuomotor interference—a higher variance on action execution when participants observe incompatible vs. compatible movements—has been described (*Gandolfo et al., 2019*; *Kilner, Paulignan & Blakemore, 2003*). To date, only few studies have considered how visuomotor interference is modulated by social contexts. We recently provided evidence that *online* interference occurs when an observed movement requires an incongruent grasp with respect to the prehension *simultaneously* observed and executed (*Chinellato, Castiello & Sartori, 2015*). This result suggests that observing an interactive gesture automatically generates an internal representation of the most appropriate response. Such representation can cause interference in the simultaneous execution of a different grasping movement, due to competition between the two motor plans (i.e., *online interference*). Here, we aim to extend previous results by investigating whether visuomotor interference can modulate perception-action coupling even *after* action observation (i.e., *offline interference*).

So far, most of the studies investigating social interactions have been confined to settings where participants were instructed (or trained) to perform similar (imitative blocks) or dissimilar (complementary blocks) actions (*Ocampo & Kritikos, 2010*; *Ocampo, Painter & Kritikos, 2012*; *Newman-Norlund et al., 2007*; *Poljac, Schie & Bekkering, 2009*; *Van Schie, Van Waterschoot & Bekkering, 2008*). Here, the paradigm did not entail any imitative or complementary blocks and no instructions were given before the trials. An auditory Go

signal was released just *after* video presentation, signaling which type of response had to be executed. Notably, this procedure let participants prepare a natural motor response, that could be *incidentally* congruent or incongruent with respect to the subsequent requested gesture. When participants are instructed to actively oppose an observed action (e.g., during complementary blocks), conflict detection and response selection processes are engaged in advance of viewing the action rather than in reaction to it (*Cross & Iacoboni, 2014a*). As compared to *incidentally* matched or mismatched stimulus–response pairs, *intentional* imitation or counter-imitation widely recruit top-down cognitive control networks (*Campbell & Cunnington, 2017*) and might suppress motor resonance (*Bardi et al., 2015*; *Cross & Iacoboni, 2014b*).

In order to induce social motor priming, we capitalize here on a well-established reach-to-grasp paradigm (for a review, see *Sartori & Betti, 2015*). Kinematic analysis of reach-to-grasp tasks is more effective in detecting subtle effects in social contexts, compared to simple reaction time measurements used in classic intransitive tasks (e.g., tapping) (for a review, see *Krishnan-Barman, Forbes & Hamilton, 2017*). To induce a full range of facilitation/interference effects, we devised a full-factorial experimental design (Fig. 1). Participants observed two videos of an actress: (i) grasping a tablespoon with a precision grip (PG), pouring sugar in a mug located nearby, and then stretching out her arm trying to pour some remaining sugar into a mug located out of her reach, in the observer's direction (Interactive action; Fig. 2A), (ii) grasping a tablespoon with a PG, pouring sugar into the same first mug, and then returning to the starting point (Non-Interactive action; Fig. 2B). Notably, a number of studies has shown that this kind of interactive requests spontaneously elicits a complementary response in the observers (i.e., social affordances; e.g., *Scorolli et al., 2014*; for a review see *Sartori & Betti, 2015*; *Sartori, 2016*). Here, viewing the actress stretching out her arm without being able to pour sugar in the mug located in the video foreground—therefore closer to the participant, triggered in the participant the socially appropriate response (i.e., a whole-hand grasp -WHG- toward the mug, to grasp it and bring it closer to the actress). Participants had to observe these perceptual events, to wait for an auditory 'Go' signal, and then to either grasp a spoon with a PG (50% of trials) or a mug with a WHG (50% of trials), depending on the 'Go' signal. Two baseline conditions, in which participants simply observed a fixation cross and performed the grasping tasks (i.e., PG and WHG trials), were also set. Given that participants always observed a PG action performed by the actress (i.e., grasping the spoon), we would expect a facilitation effect when they too grasped the spoon (visuomotor priming) to the detriment of performing a different action (visuomotor interference). On the other hand, if the social request elicits the preparation of a WHG in response to what is observed, in the Interactive condition we should expect a facilitation when performing a WHG (social motor priming) with respect to a PG (social motor interference). See Fig. 1 for a schematic representation of this set of hypotheses. With this in mind, the aim of the present study was to specifically investigate the benefits and costs associated with the processing of a social request in a postponed task. In particular, we tested whether *offline* social interference only affects reaching parameters or it influences kinematics at all levels (i.e., both the reaching and grasping components). Notably, the output of the present paradigm would differ
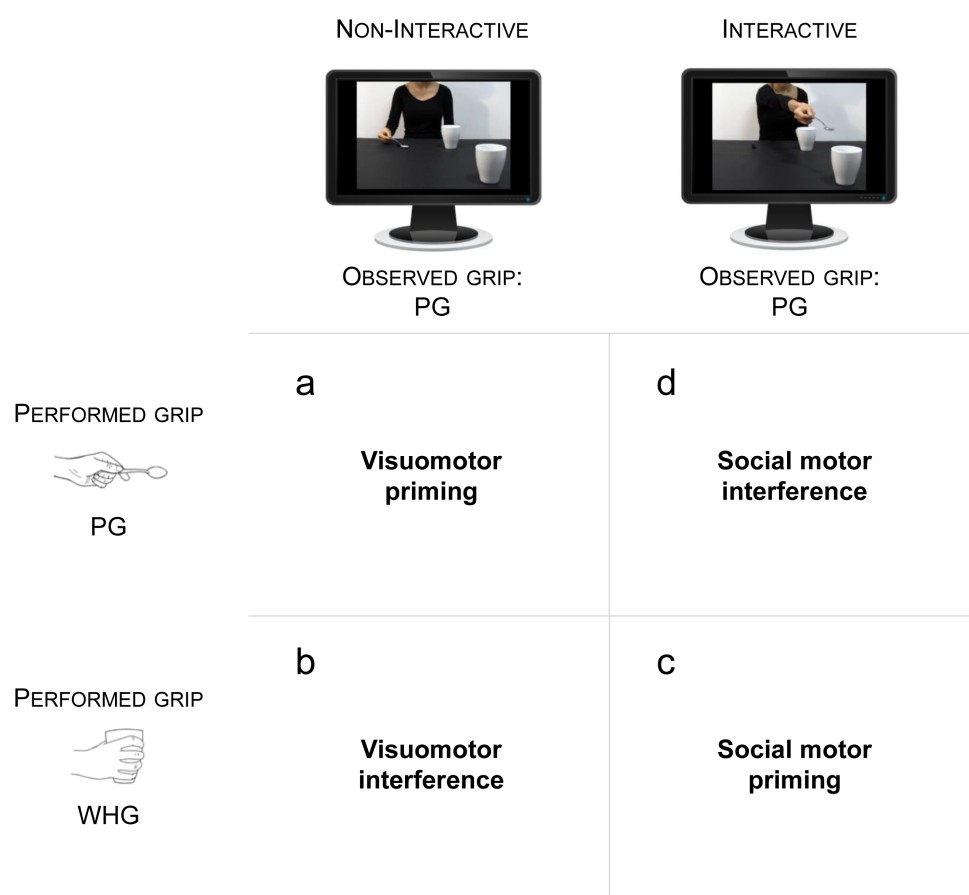

**Figure 1** **Experimental Hypotheses.** (A) Visuomotor priming is expected when the performed grip (PG) is matched with the observed non-interactive action. (B) Visuomotor interference is expected when the performed grip (WHG) is mismatched with the observed non-interactive action. (C) Social motor priming is expected when the performed grip (WHG) is matched with the social request directed to the mug. (D) Social motor interference is expected when the performed grip (PG) is mismatched with the social request directed to the mug.

from the classic Stimulus-Response Compatibility effect (*Fitts & Seeger, 1953*) in which participants are instructed to perform a finger tapping in response to observing a finger tapping (compatible) or a finger lifting (incompatible). Here, no instruction was given before the trial in terms of motor preparation, and the observed grasp was always a PG.

# MATERIALS & METHODS

## Participants

Sixteen right-handed volunteers (10 females and 6 males, between the ages of 21 and 31) participated in the experiment. A right-handed non-professional actor (female, 28 years old) was recruited for video-clips recording. All participants gave their informed written consent to participate in the study. The experimental procedures were approved by the Ethical Committee for the Psychological Research of the University of Padova by written consent (Ref. 2371) and were in accordance with the Declaration of Helsinki.
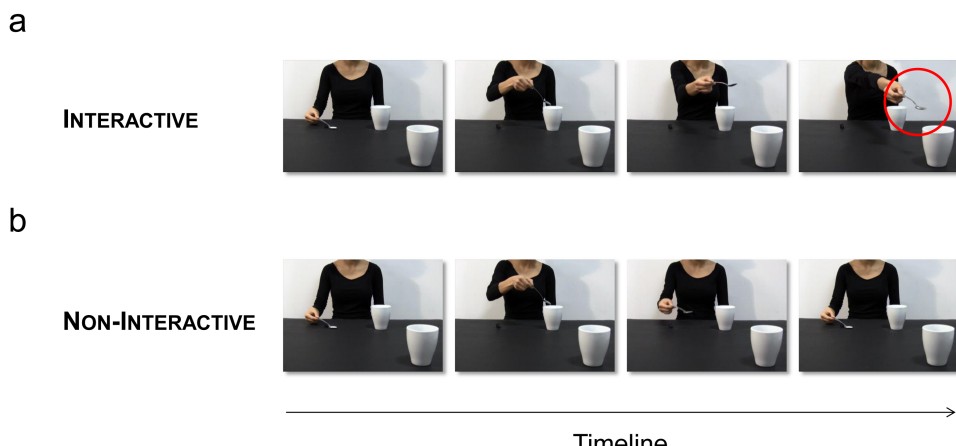

a

**INTERACTIVE**

b

**NON-INTERACTIVE**

Timeline

**Figure 2  Experimental stimuli.** (A) Interactive action: The actress pours sugar with a tablespoon (precision grip; PG) in a mug located nearby, and then stretches out her arm trying to pour some sugar into a mug located out of her reach (red circle). Crucially, this mug is placed in the video foreground, thus requiring the observer's intervention to bring the mug closer. (B) Non-Interactive action: The actress pours sugar in the same mug, and then comes back to the starting point.

## Stimuli

Two video-clips showed an actress: (i) pouring sugar with a tablespoon (grasped with a PG) in a mug located nearby, and then stretching out her arm in an attempt to pour the sugar left in the tablespoon within a mug located out of her reach (Interactive action; Fig. 2A), (ii) pouring sugar in the same mug, and then coming back to the starting point (Non-Interactive action; Fig. 2B). Crucially, the out-of-reach mug was placed in the video foreground, closer to the participant watching the video, thus eliciting a complementary reaction with a WHG when the actress was trying to reach for it. The mug was visible in the video foreground also when the actress was coming back to the starting point (Non-Interactive action), therefore controlling for possible affordance effects. In order to assess that the joint action goal (i.e., trying to pour the sugar in the mug) was clear to the participant, we performed a preliminary experiment. 50 participants (39 females and 11 males, age range 24–41) with the same characteristics of those participating in the main experiment took part in this study. They were shown the interactive video-clip and they were asked to indicate: (i) whether they felt involved in the action and (ii) what was the meaning of the actress's gesture. We avoided any given option in order to elicit a more spontaneous response. Results showed that 48 participants out of 50 (96%) reported that they felt involved in the action and 46 (92%) reported that the actress was 'asking' for the out-of-reach mug. Four (8%) participants declared that the actress was 'indicating' the mug. When asked what they had performed, they all (100%) declared that they would have grabbed and lifted the mug.

All of the videos were taken from a frontal view and were equal in length (8.2 s). Since gaze is a crucial component of social interactions and could have biased the results, the actress's face was not visible. For the participants' prehension task we adopted a sugar spoon (130 mm length, the same sugar spoon observed in the videos) vertically inserted

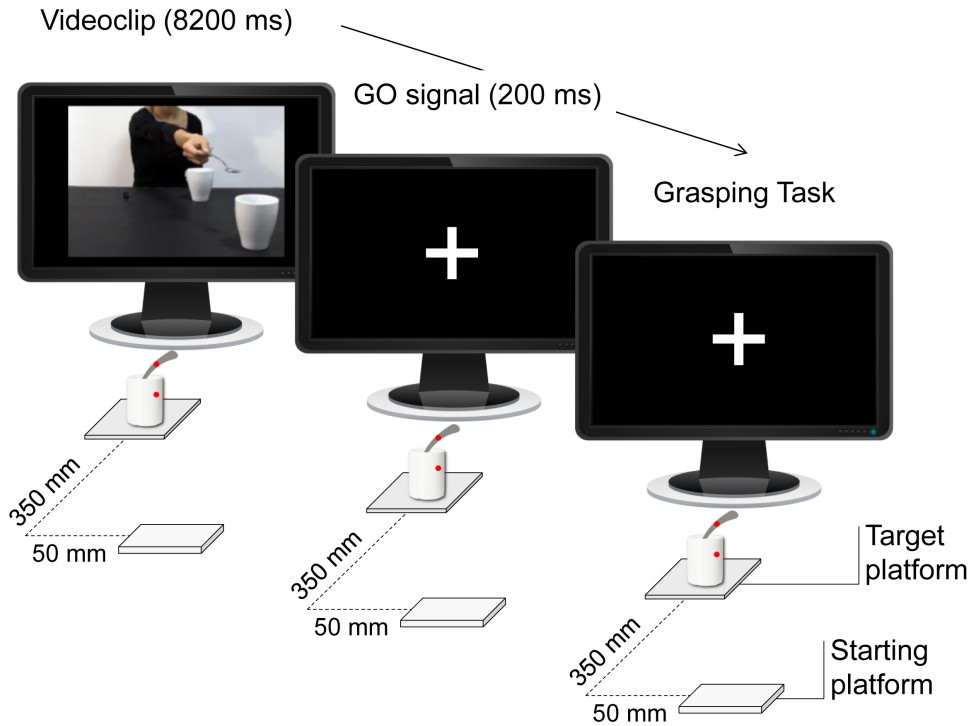

**Figure 3** **Timeline.** During the Interactive condition, the participants observed a video showing an actress stretching out her arm trying to pour some sugar into a mug located close to them, thereby inviting them to grasp it. Depending on the specific type of 'Go' signal, they then performed a reach-to-grasp task on the mug or on the spoon located on a target platform.

into a mug (90 mm diameter, the same mug observed in the videos). An affixed colored dot on the sugar spoon and on the mug was signaling the required thumb's contact-point to perform stable and consistent grasps across the experiment and across participants. Two auditory signals (a low-pitch tone, 300 Hz, 200 ms; and a high-pitch tone, 500 Hz, 200 ms) were adopted as 'Go' signals at the presentation of a white fixation cross, which lasted until the end of the trial (Fig. 3).

## Procedure

The experimental set up is depicted in Fig. 4. Participants sat on a chair in front of a table (900 × 900 mm), watched the videos that were presented on a 19″monitor (resolution 12,80 × 1,024 pixels, refresh frequency 75 Hz, background luminance of 0.5 cd/m²) set at eye level (the eye-screen distance was 60 cm). A starting platform (60 × 70 mm; five mm thick) was attached 90 mm away from the table surface's edge and 50 mm away from the midsection. After video presentation, participants had to execute a reach-to-grasp movement towards either a spoon or a mug placed on a target platform (100 × 100 mm; five mm thick), located 350 mm from the starting platform. The experiment included six experimental conditions; notably, the observed grasp was always a PG:

- Interactive action, Performed PG (Interactive PG): participants performed a PG after observing the Interactive request toward the mug.

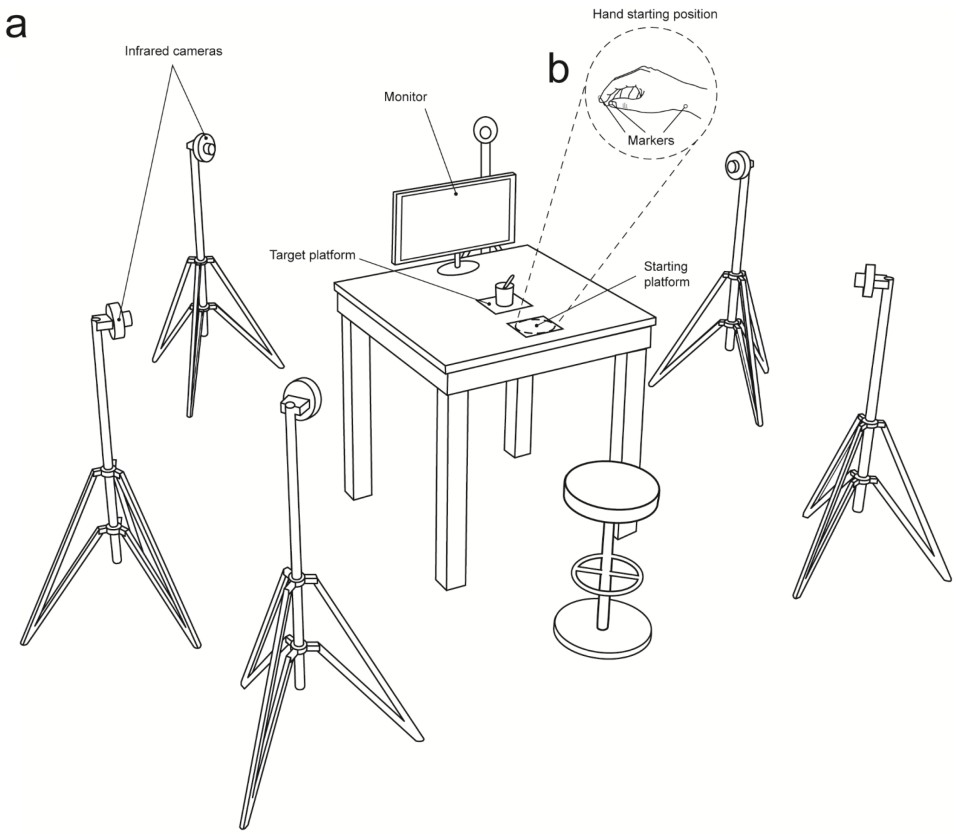

**Figure 4** **Set up.** (A) A 3D Optoelectronic SMART-D system was used to track the kinematics of the participant's right upper limb by means of six infrared cameras equipped with highly sensitive CCD sensors. Each participant sat in front of a table and had to watch the video clips (Interactive, Non-Interactive) that were presented on a monitor. (B) The participant's right elbow and wrist were resting on the table surface with the hand resting on a starting platform. To measure the grasp and reach components of the movement, retro reflecting markers were taped to the following points: thumb, index finger, and wrist.

- Interactive action, Performed WHG (Interactive WHG): participants performed a WHG after observing the Interactive request toward the mug.
- Non-Interactive action, Performed PG (Non-Interactive PG): participants performed a PG after observing the Non-Interactive action.
- Non-Interactive action, Performed WHG (Non-Interactive WHG): participants performed a WHG after observing the Non-Interactive action.
- Baseline PG: participants performed a PG on the sugar spoon after observing a white cross presented at the center of the monitor for 6 s.
- Baseline WHG: participants performed a WHG on the mug after observing a white cross presented at the center of the monitor for 6 s.

The baseline conditions were performed before the experimental session to allow participants to familiarize with the 'Go' signals and to provide baseline data for both types of grasp. The experiment was composed of 60 trials (10 per condition, each lasting 11 s). The acoustic 'Go' signal was released at the offset of each video or at the offset of the fixation

cross in the baseline conditions. Participants were instructed to begin their movements as soon as the 'Go' signal sounded and to perform either a PG or a WHG. Trials were presented in randomized order and the association between required type of grasp and corresponding auditory signal was counterbalanced across participants. The time interval between the end of the video and the presentation of the 'Go' signal was varied randomly to reduce rhythmical effects (1200-2400 ms range).

## Kinematics recording

A 3D Optoelectronic SMART-D system (Bioengineering Technology and Systems, B |T |S |) was used to track the kinematics of the participant's right upper limb. Six digital infrared cameras (sampling rate 60 Hz) equipped with highly sensitive CCD sensors were placed in a semicircle at 1–1.2 m from the table (Fig. 4A). The spatial resolution of the recording system was 0.3 mm over the field of view. Two reflective markers (0.25 mm in diameter) were placed on each participant's hand to measure the grasping component of the action (i.e., concerning finger pre-shaping and finger closing around the object), and one marker was placed on the wrist to measure the reaching component of the action (i.e., concerning hand transportation toward the target object). In particular, the three infrared reflective markers were taped to the following points: thumb (ulnar side of the nail), index finger (radial side of the nail), and wrist (dorsodistal aspect of the radial styloid process) (see Fig. 4B). Following data collection, the SMART-D Tracker software package (B |T |S |) was used to provide a 3D reconstruction of the markers' positions as a function of time.

## Data analysis

The temporal delay between the 'Go' signal and movement onset (i.e., the time at which the tangential velocity of the wrist marker crossed a threshold of five mm/s and remained above it for longer than 500 ms) was computed as Reaction Time (RT). Movement Time (MT) was then computed as the time interval between reaching onset and end of grasping (i.e., the time at which the hand opening velocity crossed a threshold of five mm/s after reaching its minimum value and remained above it for longer than 500 ms). Further, the maximum distance reached by the 3D coordinates of the thumb and index finger (Maximum Grip Aperture, MGA) was extracted for each individual movement.

The mean values for each parameter of interest were determined for each participant and entered into repeated-measures $3 \times 2$ ANOVAs with Condition (Interactive, Non-Interactive, Baseline) and Type of grasp (PG, WHG) as within-subject factors. Preliminary analyses were conducted to check for normality, sphericity, univariate and multivariate outliers, with no violations noted. Bonferroni correction was applied and a significance threshold level of $p < 0.05$ was set for all statistical analysis.

## RESULTS

A significant interaction of Condition by Type of Grasp was shown for RTs ($F_{(2,30)} = 50.341$, $p < 0.001$, $\eta_p^2 = 0.770$) and for MT [$F_{(2,30)} = 30.335$, $p < 0.001$, $\eta_p^2 = 0.669$]. For MGA, a significant effect of Type of Grasp ($F_{(2,30)} = 12.292$, $p = 0.003$, $\eta_p^2 = 0.450$) and a significant interaction of Condition by Type of Grasp ($F_{(2,30)} = 5.872$, $p = 0.007$, $\eta_p^2 = 0.281$) emerged.

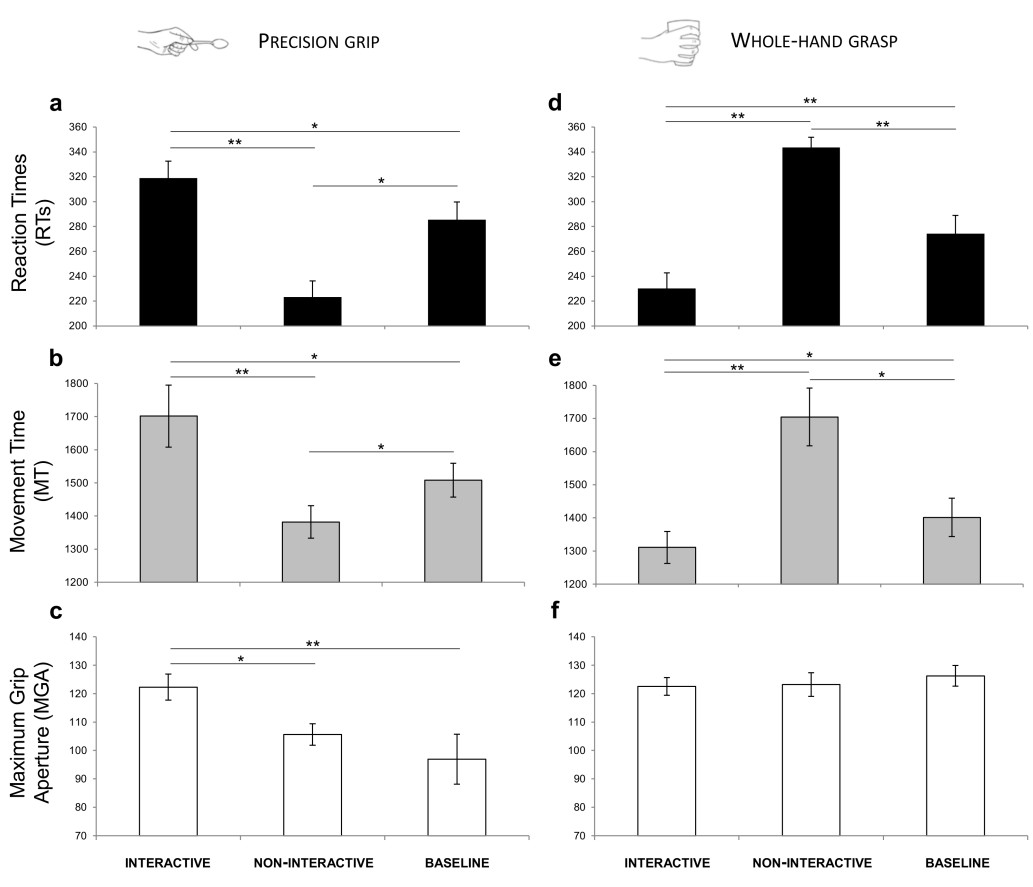

**Figure 5** **Results.** Graphical representation of the mean values for RTs (black; A, D), MT (gray; B, E) and MGA (white; C, F) across experimental conditions (Interactive, Non-Interactive, Baseline) when participants either performed a PG (A–C) or a WHG (D–F). Bars represent standard error of the mean. Asterisks indicate statistically significant comparisons, (*) $p < 0.05$, (**) $p < 0.01$.

The results obtained from the post-hoc contrasts exploring the interactions are graphically represented in Fig. 5 and listed according to the main hypotheses:

## Visuomotor priming
- RTs. Decreased RTs were found for the Non-Interactive PG compared to the Baseline PG condition ($p = 0.019$, $\eta_p^2 = 0.400$; Fig. 5A).
- MT. Decreased MT was found for the Non-Interactive PG compared to the Baseline PG condition ($p = 0.015$, $\eta_p^2 = 0.419$; Fig. 5B).
- MGA. No significant effect was found for the Non-Interactive PG with respect to the Baseline PG condition ($p = 1.00$, $\eta_p^2 = 0.041$; Fig. 5C).

## Visuomotor interference
- RTs. Increased RTs were found for the Non-Interactive WHG compared to the Baseline WHG condition ($p = 0.008$, $\eta_p^2 = 0.466$; Fig. 5D).
- MT. Increased MT was found for the Non-Interactive WHG compared to the Baseline WHG condition ($p = 0.038$, $\eta_p^2 = 0.348$; Fig. 5E).

- MGA. No significant effect was found for the Non-Interactive WHG with respect to the Baseline WHG condition ($p = 0.781$, $\eta_p^2 = 0.084$; Fig. 5F).

### Social motor priming

- RTs. Decreased RTs were found for the Interactive WHG compared to the Non-Interactive WHG condition ($p < 0.001$, $\eta_p^2 = 0.780$; Fig. 5D) and to Baseline WHG values ($p = 0.002$, $\eta_p^2 = 0.537$; Fig. 5D).
- MT. Decreased MT was found for the Interactive WHG compared to the Non-Interactive WHG condition ($p = 0.005$, $\eta_p^2 = 0.499$) and to Baseline WHG values ($p = 0.036$, $\eta_p^2 = 0.353$; Fig. 5E).
- MGA. No significant effect was found for the Interactive WHG with respect to Non-Interactive WHG ($p = 1.00$, $\eta_p^2 = 0.005$) and to Baseline WHG values ($p = 0.633$, $\eta_p^2 = 0.102$; Fig. 5F).

### Social motor interference

- RTs. Increased RTs were found for the Interactive PG compared to the Non-Interactive PG condition ($p < 0.001$, $\eta_p^2 = 0.689$) and to Baseline PG values ($p = 0.021$, $\eta_p^2 = 0.393$; Fig. 5A).
- MT. Increased MT was found for the Interactive PG compared to the Non-Interactive PG condition ($p = 0.001$, $\eta_p^2 = 0.584$) and to Baseline PG values ($p = 0.021$, $\eta_p^2 = 0.395$; Fig. 5B).
- MGA. A significant effect was found for MGA. In particular, grip aperture was increased when performing a PG for the Interactive condition compared to the Non-Interactive condition ($p = 0.038$, $\eta_p^2 = 0.349$) and to Baseline PG values ($p = 0.008$, $\eta_p^2 = 0.463$; Fig. 5C).

## DISCUSSION

This study aimed to investigate social motor priming effects in a postponed task (i.e., offline). Our findings confirmed earlier research by showing that action observation facilitates congruent types of actions (visuomotor priming) and interferes with different types of actions (visuomotor interference). At the same time, our results extend previous literature on social interactions by showing that interactive requests can facilitate incongruent—but appropriate—responses (i.e., social motor priming) and interfere with congruent—but inappropriate—grasping actions (i.e., social motor interference). Notably, this effect spontaneously occurs even in a postponed task.

### Social vs. non-social motor interference

Do visuomotor and social motor interference involve overlapping or distinct mechanisms? Interestingly, we found a crucial dissociation. Social motor interference inflicted a more serious cost on kinematics than visuomotor interference. MGA was significantly increased when participants performed a PG after having (spontaneously) planned a WHG. No modulation of the grasping component was found in the case of visuomotor interference. Given that grip amplitude covaries linearly with object size (Jakobson & Goodale, 1991),

the change in MGA was seemingly due to the previously observed social request. This output (i.e., an increased hand aperture) possibly indicates the integration of another motor plan (i.e., a WHG) when performing the offline precision grip. The present data take previous results a step further by demonstrating that social motor priming—once spontaneously triggered—is rather impervious to be inhibited even during postponed action execution. In fact, it affects both the reaching and the grasping components of performed actions. This might ultimately suggest that social motor priming is more pervasive than visuomotor priming. Notably, shifting from imitating other's action to representing the most appropriate complementary response occurs very quickly (i.e., functional shift; *Sartori, Bucchioni & Castiello, 2013*). Here we suggest instead that shifting from preparing a complementary response to mirroring is very expensive in terms of motor interference. Is this due to the intrinsically social valence of the stimulus? In this regard, we recently provided evidence that activity in the primary motor cortex elicited by a complementary request is even impermeable to attention-diverting cues (*Betti et al., 2017*). Social motor preparation seems therefore to be a genuine automatic (i.e., not reprogrammable) mechanism.

## Integration as a byproduct of offline social motor interference

Previous reports on simultaneous observation-execution interference tasks (i.e., online interference; *Chinellato, Castiello & Sartori, 2015*) indicated that planning a socially-appropriate WHG had a repulsive effect on what was performed (i.e., decreasing the MGA of the PG). This effect was likely driven by a form of inhibition of the features shared by perception and action (for a review see *Castiello, 1999*). According to *Schubö, Aschersleben & Prinz (2001)*, the representations that underlie different activities, such as producing a movement while simultaneously coding a perceptual event, must be kept distinct so that the two activities can be carried out without interfering (*Schubö, Aschersleben & Prinz, 2001*). Rather, our results for the postponed task (i.e., *offline* interference) suggest an *integration* of the inhibited motor plan (i.e., a WHG) into the executed action.

In future studies, the adoption of real-life meaningful interactions involving a set of different potential actions (e.g., *Becchio et al., 2008*) will permit to ascertain whether the integration documented here also generalize to more complex contexts and situations. For instance, dynamic interactions will allow testing pair's mutual compensation when the poor performance of one individual is fixed at the dyadic level (*Era et al., 2018b*).

## The continuum hypothesis

In terms of online interference, participants' motor performance is less affected when they have to interact in a joint task with respect to an isolated context (*Sacheli, Arcangeli & Paulesu, 2018*). In other circumstances, however, the interference due to physical incongruence is only partially decreased (*Clarke et al., 2018*)—or even increased (*Della Gatta et al., 2017*)—by the joint goal. For instance, when participants have to draw circles or lines with their right hands while observing on the screen the circles or lines that are simultaneously drawn by a partner, this leads to an increase—rather than a decrease—in interference effects during the Joint Action condition. Based on these findings, we

hypothesize that if a joint action is well consolidated in memory, then less resources are needed to process it, and therefore less interference occurs. Likewise, less interference occurs for a "second task" if the primary task is automatized (*Castiello & Umiltà, 1987*; *Castiello, 1996*; *Guillery, Mouraux & Thonnard, 2013*). Indeed, while the task in *Sacheli, Arcangeli & Paulesu (2018)* was well learnt after a training phase of 20 min, *Clarke et al. (2018)* participants performed only four trials before each block, and no training at all was performed by *Della Gatta et al. (2017)*. When a particular stimulus–response pair is well learnt, a reduction in top-down control would indicate its automaticity over other less frequently experienced responses (*Campbell & Cunnington, 2017*). Altogether, these findings raise the possibility that motor interference is an *experience-based continuum*, rather than an on/off mechanism. This hypothesis would also explain why the life-long learnt response of hand shacking is so automatic (*Liepelt, Prinz & Brass, 2010*). On the contrary, one might argue that interference is a simple on/off effect produced by the online competition between different simultaneous representations. In that case, motor interference should vanish in a postponed task since competition would be no longer active. But in the present experiment this was not the case. Our findings extend previous research by revealing that physically incongruent action representations can be integrated into a single action plan even during an offline task and without any training. Notably, our paradigm did not entail any imitative or complementary blocks and no instructions regarding the action to be performed were given before trials began, as it occurred in other previous studies (*Newman-Norlund et al., 2007*; *Van Schie, Van Waterschoot & Bekkering, 2008*; *Poljac, Schie & Bekkering, 2009*; *Ocampo & Kritikos, 2010*; *Ocampo, Painter & Kritikos, 2012*). This approach was adopted in order to investigate a purely spontaneous phenomenon, rather than a learnt effect. This aspect is important as in real life many cooperative and competitive contexts require partners to flexibly adapt their responses to others' actions, without the chance to previously know or even practice the specific response that they must implement (e.g., *Becchio et al., 2008*; *Sacheli et al., 2015*).

## Social response: low-level or high-level mapping?

According to the Social Associative Memory hypothesis, an associative mechanism would be in charge of matching certain actions to their natural social response, irrespective of who is actually performing the action (*Chinellato et al., 2013*; *Chinellato, Castiello & Sartori, 2015*). "If action B (e.g., take) usually follows action A (e.g., give), the observation of a partner executing A elicits the pre-planning of B by the observer. On the other hand, if the subject executes A, he expects to see the partner performing B in response" (*Chinellato et al., 2013*). Here, extensive experience of carrying out complementary actions in a social context would result in automatically generating the complementary action when observing an action in a social context. Consistent literature on social Simon effects (*Guagnano, Rusconi & Umiltà, 2010*; *Humphreys & Bedford, 2011*; *Dittrich et al., 2013*) and the development of Stimulus-Response associations (*Catmur, Walsh & Heyes, 2009*) provides convergent data on the hypothesis of a low-level direct mechanism for the priming of different behaviors. Our results, based on a *postponed* interference effect, might indeed be attributable to general processes of associative learning (*Catmur, Walsh & Heyes, 2009*; *Massen & Prinz, 2009*).

An alternative account for the present data is the high-level mapping. The theory of event coding (TEC; *Hommel & Elsner, 2009*; *Hommel, 2009*) states that observed (and executed) actions are represented in the form of their distal consequences. The TEC is based on the common coding hypothesis, which claims that perception and action rely on shared cognitive representations. According to the TEC, translating a perceived human movement into corresponding motor programs would function as an emulator, tracking the behavior of conspecifics in real time to generate predictions of an unfolding action (*Wilson & Knoblich, 2005*). Our data, on the contrary, show that observing others' behaviors rapidly activates appropriate complementary motor plans in an observer. In fact, since the distal goal of the actor is to reach the distant cup and the most efficient action to do it (by herself) would be slightly rising from her seat, motor prediction should have activated in the observer the corresponding leg muscles, rather than right-hand muscles. On the other hand, it is plausible that both a predictive and a social motor response preparation might have taken place, as we recently demonstrated (*Sartori et al., 2015*). In conclusion, the reported effects are an example of a spontaneous tendency to fulfill the request embedded in a social interaction. This might be confirmed by the fact that a control group reported that they were ready to lift the salient object toward the model. It could be argued that attention played a role in modulating motor priming and that the actor's hand—moving toward the object—was simply more salient then the hand moving back to the starting position, without the effect being intrinsically a social motor priming. If this were the case, then a simple arrow presented instead of the hand would have produced similar findings. However, results from previous studies in which the social request was substituted by an arrow did not provide support for this view (*Flach et al., 2010*; *Sartori et al., 2011*). Rather, we suggest that the motor system is preferentially tuned to meaningful actions of interactive partners.

## CONCLUSIONS

The purpose of this study was to investigate whether visuomotor interference can modulate perception-action coupling even after action observation (i.e., *offline* interference). The present results suggest that physically incongruent action representations can be integrated into a single action plan even during an offline task and without any training. The future goal is to design stringent paradigms that might allow to compare findings from real-life meaningful interactions involving a set of different potential actions.

### Funding

This work was supported by a Scientific Independence of Young Researchers grant (SIR - N. RBSI141QKX) to Luisa Sartori and by Progetto Strategico, Universita' di Padova (N. 2010XPMFW4) to Umberto Castiello. The funders had no role in study design, data collection and analysis, decision to publish, or preparation of the manuscript.

## Grant Disclosures

The following grant information was disclosed by the authors:

Scientific Independence of Young Researchers grant: SIR - N. RBSI141QKX.

Luisa Sartori and by Progetto Strategico, Universita' di Padova: 2010XPMFW4.

## Competing Interests

The authors declare there are no competing interests.

## Author Contributions

- Sonia Betti conceived and designed the experiments, prepared figures and/or tables, authored or reviewed drafts of the paper, approved the final draft.
- Eris Chinellato analyzed the data.
- Silvia Guerra performed the experiments, prepared figures and/or tables.
- Umberto Castiello conceived and designed the experiments, authored or reviewed drafts of the paper, approved the final draft.
- Luisa Sartori conceived and designed the experiments, performed the experiments, contributed reagents/materials/analysis tools, prepared figures and/or tables, authored or reviewed drafts of the paper, approved the final draft.

## Human Ethics

The following information was supplied relating to ethical approvals (i.e., approving body and any reference numbers):

The project has been approved by the Ethical Committee for the Psychological Research of the University of Padova by written consent (Ref. 2371).

## Data Availability

All the raw kinematic data are available in the Supplemental File.

## Supplemental Information

Supplemental information for this article can be found online at http://dx.doi.org/10.7717/peerj.7796#supplemental-information.

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
