# Peer review of "Social Motor Priming: when offline interference facilitates motor execution"

_PeerJ, doi:10.7717/peerj.7796_

## Round 0.1 · original submission · Major Revisions

I have now received the comments of two reviewers, experts in the field, on the manuscript that you submitted to PeerJ, and have read your manuscript myself. I thank the reviewers for their work. The reviews are very thoughtful and quite clear, so please refer to them for details.

I see two major points that you will need to address for a successful revision: a. Reviewer 1 argues that your context is not really a social/ interactive one (see also comment of reviewer 2). You should address this concern, describing potential limitations of the paradigm you chose; b. better highlight the novelty of your results with respect to current studies (see comments of Reviewer 2). You will also need to better ground your work in current literature; both reviewers offer excellent suggestions; you might want to add Scorolli et al. 2014, Neuropsychol. for a very similar task.

Both reviewers converge on a major revision verdict, and I agree with their assessment.

Thank you for sending your interesting work to PeerJ.

Reviewer 1 ·

Basic reporting

This is a reasonably executed paper. However, the theoretical motivation is unclear. Specifically, is it joint action, or social context? The authors mention both but it’s not clear if they intend the terms to be interchangeable. If not, these are two separate mechanisms, and their influence and interaction needs to be clarified.

Experimental design

I think my main issue is that I am not convinced this is actually a social/ interactive context. Participants were watching the model on the video and making same / different actions, as in previous literature. They were not 'sharing' objects, however. Moreover, according to the stated methods, participants were never given any instruction that might bias them towards feeling an association (identification with) the model . Also, as far as I can tell participants also were not led to believe that the mug was ‘theirs’ in any way, so did they really believe they were engaging in a social event with the model?
If social context is not at play here, then I’m unsure how the findings advance on Newman- Norlund et al, and on Ocampo et al.

I would recommend another experiment, in which participants were given actual 'social' instruction

Validity of the findings

The section labelled ‘Visuomotor Interference’ is, I think, over-interpreted. It is difficult to conclude this for comparisons with Baseline, unless the participants perform the action in the presence of some other salient non-movement stimulus, yielding no interference. That is, it could just be the significantly 'large' amount of info that is causing disruption. Eg, a still image of the model + mug, that has been scrambled

Additional comments

Minor points:
-the paper would benefit from a good editing pass, from a native English speaker. Sentences such as line 47-49: “Visuomotor priming, in particular, regards the facilitation to perform an action congruent with the observed one versus the difficulty to execute the same action during the observation of a different one” and line 72-74 “On the other hand, evidence that online interference occurs when an given action has to be performed, but a concurrently observed movement elicits a complementary different grasp has been provided using a reach-to-grasp task.” are hard to follow.
-not clear how the idea of a continuum of interference was deduced - is it the magnitude of kinematic differences? I think the authors simply mean it can arise due to different mechanisms
-other relevant social context literature should be included: S. Sparks, T. Douglas & A. Kritikos, QJEP, 2016; Leighton, J. Bird, G., Orsini, C., & Heyes, C. (2010). Journal of Experimental Social Psychology,

Reviewer 2 ·

Basic reporting

The authors report the results of one experiment in which they ask participants to perform either precision or whole-hand grips, after observing video of an actor performing precision grips in an interactive and non-interactive context, or without observing any action before (baseline). This experimental design allowed the authors to investigate: visuomotor interference, visuomotor priming, social motor priming and social motor interference. In the Interactive condition the socially appropriate response was a complementary action (performing a whole-hand grip). Results show that while visuo-motor interference effect was only present in the Non-Interactive condition, in the Interactive condition, incongruent (yet socially complementary and appropriate) actions were facilitated with respect to congruent ones.
Although not so novel, in light of previous studies investigating similar processes, the results of the present study are interesting and the study is well designed. Moreover, the article is well written and the images are very well done. I would suggest to better explain the novelty of the present findings and to clarify the hypotheses tested in the present study. Moreover I suggest to expand the introduction and better explain some methodological aspects as highlighted in the following points.

In the introduction, lines 89 – 94, the authors claim that “Another possibility is that a simple effect of co-representation might explain the interference effect. In that case, motor impairment would not last in a postponed task.” Could you please explain why the motor impairment should not last in a postponed task? This seems an important issue, given that this statement apparently sets the ground for your hypothesis. In fact, in lines 96-97, you write that “the present study has been designed to specifically test whether social compatibility effects are long-lasting and can modulate the perception-action coupling even after action observation.” I find this connection not very straightforward. It seems that you are implying that finding a delayed social compatibility effect would support the “continuum” hypothesis and disprove the “co-representation” one? If so, why and how? Please explain.

In the introduction the authors refer to studies reporting reduced visuo-motor interference when participants are acting towards reaching a joint goal. Please also discuss other studies indicating that during joint-actions performing complementary actions is not more difficult than performing imitative ones (Sacheli et al., 2015 Nat Commun, Sci Rep; 2018 JCN; Era et al., 2018 SCAN, Psych Res; Gandolfo et al., 2019 Acta) and suggesting that when interacting with a partner the need to predict the other’s actions and to program one’s own action accordingly might abolish visuo-motor interference, that is instead present when acting at the same time without needing to predict what the other is doing.

Experimental design

If the authors are interested in creating an interactive scenario, why was the action participants needed to perform (precision or whole-hand grip) pre-instructed? Participants could have been asked to complement or to imitate the action of the actor, without knowing in advance what action to perform (maybe adding more actions to possibly perform, to create more variability) and thus creating a more naturalistic scenario. Please comment.

In the data handling and results section Accuracy is not reported (number of correct grasping). Please report this measure.

Validity of the findings

Please highlight the novelty of your findings with respect to previous research. Are they new because they demonstrate that social interference also occur in a postponed task? Please elaborate more on this point.

In the Introduction (lines 88-90) you suggest that motor interference might be described as a continuum.
In Conclusion (lines 298-299) you write that “It seems therefore that joint action goals reduce visuomotor interference effects along a continuum”. However, it is not clear which of the reported results would support this conclusion. Please explain.

Also, in lines 296-298, you write that “interference due to observing a precision grip and subsequently perform a whole-hand grasp was partially reduced for the interactive compared to the non-interactive context (i.e., quicker RT and movement time)”. However, from the results it appears that interference effect is abolished in the interactive condition and not just partially reduced. Please explain.

Looking at the plot in Fig. 5, it seems that the mean MGA value for PG in Interactive condition is almost comparable to the MGA values for WHG. This seems quite odd (and potentially interesting), as it would mean that participant’s grip aperture in Interactive-PG was as large as when performing a WHG. Also related to the previous point about the Accuracy, can you confirm that only correct trials (trials in which participants were performing PG are required by the instruction) were entered in the analysis? Please comment.

---

## Round 0.2 · Minor Revisions

I am happy to accept your paper, I would only ask to proofcheck it once again (see comments of Reviewer 1).

Thank you

Reviewer 1 ·

Basic reporting

The literature review and theoretical context is now much improved

Experimental design

While there are still limitations to the original design, the authors have addressed and discussed these adequately.

Validity of the findings

no comment

Additional comments

I would recommend another editing pass for the new sections by a native English speaker.

Reviewer 2 ·

Basic reporting

No comment

Experimental design

No comment

Validity of the findings

No comment

Additional comments

I am happy with the work made by the authors.

---

## Round 0.3 · accepted · Accept

I am happy to inform you that your paper has been accepted for publication in PeerJ. I think it represents a strong and novel contribution to the field.